# Decisions of Greek Courts Securing the Right of Parent–Child Communication and Their Determinants

**DOI:** 10.3390/healthcare10122522

**Published:** 2022-12-13

**Authors:** Andreas-Nikolaos Koukoulis, Maria Tsellou, Vasiliki Rougkala, Flora Bacopoulou, Stavroula Papadodima

**Affiliations:** 1Department of Law, School of Law, Democritus University of Thrace, 691 00 Komotini, Greece; 2Department of Forensic Medicine and Toxicology, School of Medicine, National and Kapodistrian University of Athens, 115 27 Athens, Greece; 3Department of Management Science and Technology, School of Economics and Business, University of Patras, 263 34 Patra, Greece; 4Center for Adolescent Medicine and UNESCO Chair on Adolescent Health Care, First Department of Pediatrics, School of Medicine, National and Kapodistrian University of Athens, 115 27 Athens, Greece; 5Aghia Sophia Children’s Hospital, 115 27 Athens, Greece

**Keywords:** parent–child communication, parental alienation, child abuse, interference, Greece, legal database

## Abstract

Background: There is an increasing awareness that a child’s separation from one parent after the divorce places the child’s development and well-being at risk. The aim of this study was to determine how Greek courts deal with the cases of parental prevention of communication with their children and which factors affect the judicial decisions. Methods: The Greek legal databases “NOMOS” and “Isokratis” were searched, and associations between judicial decisions, as well as communication prevention ways, and several parameters, were assessed. Results/Conclusions: A total of 50 parental communication prevention law cases were retrieved for the time period from 1992 to 2019. Results showed that mothers were more frequently alleged to interfere with father–child communication. Both direct and indirect methods of interfering with communication were followed. In cases of a single child, the method of indirect interference was more frequently followed. Judicial decisions were unaffected by the age and gender of the child, the gender of the parent preventing the communication, the number of children, the gender of the child and whether the child is the same gender as the preventing or prevented parent, the way of prevention, and the reference to parental alienation.

## 1. Introduction

The phenomenon of a child’s strident rejection of one parent, generally accompanied by strong resistance or refusal to visit or have any contact with that parent after parental divorce, has increasingly troubled and gained a great deal of attention from family courts, professionals, and parents. The term “parental alienation syndrome” (PAS) was proposed by Gardner to describe a psychiatric disorder in a child that arises almost exclusively in the context of a custody dispute [1]. Since then, the above syndrome has been debated and has received extended critique as not adequately scientifically sound. At the same time, there is an increasing awareness that separation from one parent places the development and well-being of children at risk. Furthermore, the term “parental alienation” appears to be frequently used in cases of alleged child abuse and psychological tools have been considered inadequate to discriminate parental alienation, child abuse, or bad parenting [2].

Divorce has become a significant social phenomenon in Greece with a 10 year rise of 45.8% from 2008 to 2017. In 2017, the number of divorces amounted to 19,190 compared to 11,013 in 2016, recording an increase of 74.2%. The crude marriage and divorce rates for the year 2017 were 4.7% and 1.8%, respectively. The dependent children (under 18 years) from divorced marriages amounted to 16,103 (64.3% of the total children). Regarding the type of divorce, 5325 (27.7%) were contested divorces, with a considerable number (1012) attributed to serious breakdown of the marital relationship [3]. Given the increased number of contested divorces and the number of children involved, the issue of children’s right to communicate with both parents frequently arises.

The aim of the present study was to examine how Greek courts deal with cases of prevention (interference) of parent–child communications, which factors affect judicial decisions, and whether specific arrangements are in place when child abuse accusations arise during the trial.

## 2. Materials and Methods

### 2.1. Data Search

The Greek legal databases “NOMOS” and “Isokratis” include anonymized cases and important law texts from Greek courts, usually deposited by lawyers who have dealt with them. Cases included in the above databases were chosen in an almost random way and thus are rather indicative of the situation in Greece. The law texts include final judicial decisions in which the reasons why the court reached its conclusion are also included. It is worth mentioning that Greece is a civil law country, which means that, unlike Anglo-American common law, judicial decisions are based only on enacted laws (in the form of codes or other statutes) (Greek Civil Code, Art. 1) and international law. The “NOMOS” and “Isokratis” Greek legal databases are characterized by data integrity, safety, and recoverability, thus ensuring the reliability and validity of the research. They are official legal databases, recognized for the accuracy and consistency of their data. Only authorized legal personnel have the right of data entry and the data stored are also double-checked by authorized personnel to validate their integrity and to spot input errors.

The aforementioned legal databases were searched for the time period from 1992 to 2019 using the following keywords: “communication”, “communication prevention”, “parental alienation”. These keywords were used alone or in combination. Two of the authors/researchers (ANK and VR, lawyers) performed the search in different time periods (with a time gap of 3 weeks between the searches) independently to avoid bias. There were no differences in the cases recovered.

None of the authors had participated in any way in the judicial procedure of any of the cases nor had they any knowledge of the identity of the people involved. As already mentioned, ANK and VR are lawyers, whereas SP and MT are forensic doctors. In Greece, forensic doctors examine cases of child abuse and are familiar with court issues regarding children. FB is a pediatrician specializing in adolescent medicine/pediatric–adolescent gynecology, and she is also frequently involved in cases of child abuse. Each scientist offered his/her own perspective to the research, which led to a more integrated approach.

The factors extracted from the database were the following:(1)The gender of the parents and the number and the gender of the children;(2)The judicial decision (acceptance of the prevention of communication or not);(3)The strategies followed to prevent communication, as described in the judicial decision;(4)Reference to the term “parental alienation” in the judicial decision;(5)The effects on child psychology and the ways of dealing with alienation, as described in the judicial decision.

The description of the psychological effects in the judicial decision was based on the evidence from testimony and the report of the child psychiatrist or psychologist (if performed).

The judicial decision refers only to the existence or not of the prevention of communication as child custody is decided during another trial.

### 2.2. Statistical Analysis

Absolute and relative frequencies were used for descriptive analysis. The chi-square (χ^2^), Fisher exact test, and Monte Carlo simulations were used to examine the associations between variables. The statistical significance level was set at 0.05 (5%). Statistical analysis was performed using the SPSS version 25.0 (IBM Corp. Released 2017. IBM SPSS Statistics for Windows, Version 25.0. Armonk, NY, USA: IBM Corp.).

To assess the chi-square test, six assumptions should be satisfied [4].
(i).Data in the cells were frequencies or counts of cases rather than percentages.(ii).Levels (or categories) of the variables were mutually exclusive.(iii).Each subject contributed data to one and only one cell in the χ^2^.(iv).The study groups were independent.(v).There are two variables, and both were measured as categories, usually at the nominal level. However, data could be ordinal data.(vi).The value of the cell expected should be 5 or more in at least 80% of the cells, and no cell should have an expected of less than 1.

Except for assumption (vi), all the others were satisfied in every analysis of the study. If the asymptotic test could not be used, Monte Carlo simulations were assessed [5].

## 3. Results

### 3.1. Descriptive Findings

A total of 50 parental communication prevention law cases were retrieved from the databases. All the allegations were made in the context of separated families and the children were residing mostly with the one parent. In 31 (62.0%) cases there was one child, in 18 (36.0%) cases there were two children or more, and in one (2.0%) case such information was not mentioned in the decision. Children prevented from communication were males in 16 (32.0%) cases and females in 20 (40.0%) cases. In 6 (12.0%) cases, children of both genders were prevented from communication, whereas in 8 (16.0%) cases the gender of the child was not mentioned. Concerning children’s age, in 10 (20.0%) cases the age was 6 years or less, in 20 (40.0%) cases the age was 7–12 years, and in 7 (14.0%) cases the age was greater than 12 years. Children of different age groups existed in 4 (8%) cases, whereas the age was unknown in 9 (18.0%) cases.

The mother was allegedly preventing the communication in 30 (60%) cases, and the father was allegedly preventing communication in 12 (24%) cases. In the rest of the cases, no information about the gender of the parent was available (8 cases, 16.0%).

The gender of the parent allegedly preventing the communication was similar to the gender of the child/children in 17 (34%) cases, whereas the opposite was recorded in 20 (40%) cases. In the rest of the cases, no information about the gender of the parent and/or of the child/children was available (15 cases, 30.0%).

Concerning the way of preventing communication, in 22 (44%) cases direct interference was recorded, in 16 (32%) cases indirect interference was recorded, and in 9 (18%) cases both were recorded. Such data were not available in 3 (6.0%) cases. Regarding direct interference, the following strategies were recorded: absence during the scheduled time of visit, disappearance, organization of extracurricular activities for the child, conflicts, and tension with the other parent concerning the communication. Regarding indirect interference, strategies involving the child were propagation, tension during contact in front of the child, and denial of communication by the child.

The judicial decision included the term “parental alienation” in 11 (22%) cases. In those cases, the decision was based on a child psychiatric report. Additionally, the court recognized parental communication prevention in 26 (52.0%) cases and rejected it in 24 (48.0%) cases.

The effects of the communication prevention on child psychology were mentioned in 6 cases: “Negative emotions that affect the child’s ability to formulate and express its true will”, “Difficulties with the other gender”, “Hostility towards the other parent, negative emotions including stress, anger, fear”, “Fear of disappointing one parent when communicating with the other”.

Allegations of sexual abuse had been made in two cases, both of them against the father. In the first case the court decided that child custody should be given to the mother based on the daughter’s testimony about sexual abuse, although there was a psychiatric report according to which there were no indications of sexual abuse, and, on the contrary, there were indications of parental alienation syndrome (Court of Appeal of Thessaloniki 1008/2008). In the second case, the appointed expert concluded that there was no evidence of child sexual abuse but evidence of parental alienation syndrome at an advanced stage and the court decided accordingly (Court of First Instance of Rhodes 657/2013).

### 3.2. Associations

Ways of preventing communication were statistically associated with the number of children (*p* = 0.041), the parent preventing the communication (*p* = 0.009), mention of parental alienation (*p* = 0.040), and decision (*p* = 0.028) (Table 1). More specifically, in 81.8% of the cases in which direct interference and in 46.7% of the cases in which indirect interference were reported, there was only one child. The respective percentage when both direct and indirect interference were used was 44.4%. Direct interference was more frequently used by mothers (83.3%) than fathers (16.7%), whereas indirect interference was used almost equally by mothers (46.2%) and fathers (53.8%). Both ways were exclusively used by mothers. When direct interference had been used as a way of preventing communication, in 36.4% of the cases the term “parental alienation” was mentioned, whereas the respective percentages for indirect interference and for both ways were 18.2% and 45.5%. Regarding the association between the decision and the ways of preventing communication, in 60.0% of cases with direct interference the prevention was accepted, whereas for indirect interference and for both ways the respective percentages were 16.0% and 24.0%.

On the other hand, non-significant associations were demonstrated between ways of preventing communication and age (*p* = 0.093), gender (*p* = 0.881), parental prevention from communication (*p* = 0.083), same gender as the parent preventing the communication (*p* = 0.418), and psychological effects on children (*p* = 0.615).

Decisions were statistically associated with the psychological effects on the child (*p* = 0.035). When psychological effects on the child involved internalization (feelings of inadequacy, difficulties with the other gender), the court accepted the communication prevention in 83.3% vs. 20.0% when feelings concerning relationship of the child with parents were involved (Table 2).

## 4. Discussion

In the past decade, there has been a significant increase in families experiencing issues of separation and divorce, and parental–child contact problems in particular. The study of Johnston supported children’s attitudes toward both parents after divorce being best described by a continuum from positive to negative [6]. The milder forms of parental alignment with one parent and mild rejection of the other are assumed as relatively normal, even before divorce. Differences in character, age, gender, cognitive capacity, shared interests, and parenting practices and style may result in children preferring one parent to the other, although such affinities may shift over time as the child grows older with changing developmental needs and situations. It is more unusual for children to demonstrate a clear alignment with one parent and want limited contact with the non-preferred parent after divorce. Most aligned children do not completely reject the other parent; rather, they tend to express some ambivalence toward this parent, including anger, sadness, and love. At the far end of the continuum are children who have extreme alignment with one parent, whereas they strongly and unjustifiably resist or completely refuse contact with the other parent without apparent ambivalence or guilt [7,8,9,10].

Children’s negative behavior and attitudes toward a parent are postulated to have multiple determinants. All parties are implicated in the problem—both the aligned and rejected parents—in addition to vulnerabilities within children themselves. Rejection of parents, whether the father or the mother, may relate to deficits in their parenting capacity, which worsen as the parent feels powerless due to the alliance against him/her. On the other hand, the aligned parents may contribute to alienating a child from the other parent. It has been suggested that the mother’s behavior can disrupt the father–child relationship more effectively than the father’s behavior can disrupt the mother–child relationship, probably because mothers have a more dependent bond with and better access to their children and subsequently have increased opportunity to exert influence [6].

Apart from factors related to parents (parenting practices, personality, mental health etc.) and their relationship before and after the divorce, child factors (age, cognitive capacity, temperament, vulnerability, special needs, and resilience) and the course of the adversarial process/litigation have also been recognized as dynamic contributing and facilitating factors [2].

In conclusion, it can be supported that the parent–child contact problem is a generally broader term which covers different types of parent–child relationships and behaviors, more specifically:(1)An affinity for one parent relating to age, gender, common interests, or a prolonged absence of the parent, which is considered relatively normal, sometimes even before divorce;(2)An alignment (arising from a loyalty conflict) with a parent as a coping mechanism to the parental separation and mild rejection of the other parent;(3)A realistic estrangement (justified rejection), usually in connection with interpersonal violence and/or child abuse;(4)Alienation (unjustified rejection) where one parent may influence or pressure the child into believing the other parent is bad, wrong, or dangerous, although they are not, which results in the child expressing fear, anger, resistance, or rejection toward the other parent.

In most of the cases, the abovementioned situations consist rather of a continuum without clear limits and have many grey areas, with both parents (and other parties such as siblings and others) being involved in their etiology. On the other side, it is extremely important for the protection and welfare of the child that cases of interpersonal violence and child abuse, as well as cases of alienation, be properly identified and treated [11,12].

For the above reasons, one of the greatest challenges faced by forensic psychologists, lawyers, and judges is the significant number of cases in which children reject a parent after the divorce. Given the general uncertainty in the field, as well as the uncertainty about the facts of specific cases, many parents and children involved in these cases are not well served by the family justice system. A court’s decisions may be ineffective due to the complexity of the situation and the lack of agreement between the two parties.

“Parental alienation syndrome (PAS)” or “parental alienation” have been used as terms in courts in Australia [11] and in Europe [13] in cases of alleged physical or sexual child abuse. The primary concept is that the one parent is coaching the child to make allegations of abuse to manipulate court outcomes, although the parent knows that there is no actual risk to the child. In those cases, parental alienating behaviors are usually related to mothers who have been described as “manipulative, mentally unwell, suffering from delusions, and ultimately harming their children with the intent of punishing the father” [11]. It seems, however, that in most cases parents have a genuine belief that their child is at risk and needs protection and parenting alienating behaviors occur without the intent to destroy the child’s relationship with the other parent, although these behaviors finally are responsible for or contribute to that outcome [14]. On the other side, it must be recognized that allegations of parental alienation may frequently be used by abusive fathers to cover and undermine intrafamilial violence and abuse in courts during child custody arrangements. The concept of parental alienation has been questioned because of the harmful impact it may have on the victims of domestic abuse, mostly mothers and their children [15]. According to the statement of the Experts on Discrimination and Violence against Women (EDVAW), the abuse of the term “parental alienation” and of other similar concepts and terms is discouraged given that this tactic could invoke the denial of child custody to the mother and grant it to a father accused of intimate partner violence in a manner that ignores the possible risks for the child [16].

Weighing the evidence and expert opinions to counteract the situations and reach a judicial decision aiming at the best interest of the child is an incredibly challenging and complicated procedure. Collaboration between mental health and family justice professionals to develop and implement effective interventions adapted to the needs of the specific family is of extreme importance. It is generally agreed that the great challenge in practice is to distinguish between alienation (unjustified rejection) and realistic estrangement (justified rejection owing to interpersonal violence or child abuse). However, there is a disagreement about the extent to which family courts may respond to cases of alienation and interpersonal violence as well as how they may balance the risks of failing to identify them [2,17]. Differentiation between child abuse and/or family violence and different levels of parental–child contact problems (parental alienation included) needs meticulous screening, assessment, and consideration of the behavior of parents and children, always in the context of the multiple and interrelated socio-political, familial, partner, and individual factors. Differentiation of the nature and severity of the parent–child contact problems, although most times challenging, is crucial for identifying the most appropriate intervention [18,19].

Under Greek law, responsibility for the minor, namely a child under 18 years of age, constitutes both a right and a responsibility of the parents, who, in principle, must exert it in common. Parental responsibility includes the upbringing / raising, supervision, education, and training of the child, as well as the determination of the child’s residence (Greek Civil Code, article 1510). Parental responsibility is awarded, therefore, to both parents at a child’s birth and is not withdrawn from one or both unless under exceptional circumstances. In the case of a divorce, if parents do not agree upon the way of exercising responsibility and allocation of specific rights and obligations, the court shall regulate the above matters. According to Article 1520 of the Greek Civil Code, the parent with whom the child does not reside retains the right of personal communication with him/her. Pursuant to the above article, parents do not have the right to prevent the child from communicating with his/her distant relatives in the ascending line relatives, unless there is a serious reason. Grandparents have the same independent right for personal contact with the child (grandchild), even in the event of the death of the parent who was their child. Violation of the above right incurs penalties by means of personal detention and/or fines (articles 950 and 947 of the Greek Code of Civil Procedure). In Greece, cases regarding allegations of prevention of parent–child communication reach the courts usually after action by one parent.

Under the Greek law, in the case a minor falls victim to domestic violence, if there is a demand by the other parent, a close relative of the child, the public prosecutor, or even ex officio, the court can order any proper means to avert the exercise of any form of violence on the child (Greek Law 3500/2006, article 12, 18). These means include the removal of parental responsibility from one parent in total or partly or the offender’s removal from the family residence. The court can also prohibit him/her from approaching the children’s residences or even schools. The same of course accounts for any other kind of abuse, sexual or psychological, or neglect [20]. The judicial decisions must conform with the standard of the best interest of the child as this derives from Article 3 of the United Nations Convention on the Rights of the Child [21]. The decision regarding parental responsibility is taken during another trial, simultaneously or not, with the trial for the prevention of the parent–child communication.

In the present study we (a) discussed parent–child communication problems, with emphasis on the prevention (interference) of parent–child communication and parental alienation; (b) referred to the concerns arising in the case of child abuse allegations during the trial; and (c) recorded cases of parental communication prevention reaching the Greek courts and determined their characteristics, as well as the factors that may have affected the judicial decision. A total of 50 cases were found in the Greek legal databases from 1992 to 2019. Obviously, not all the cases are included in the above databases, and even in the existing ones many details are lacking due to Personal Data Protection Act. These are two methodological limitations we should consider prior to interpreting the findings and drawing conclusions. However, even in the presence of the above limitations, we believe that these results are indicative of the situation in Greece.

In our sample, mothers allegedly prevented communication of children with their fathers more frequently (60% of the cases), which is well described in the literature. Children in litigating families have shown more evidence of alignment with their mothers and correspondingly more evidence of rejection of their fathers [6]. In their study of family law judgements in the USA, Meier and Dickson found that 82% of the alienation claims were brought by fathers. This was consistent with the fact that most parents starting with primary custody (75%) were mothers [22]. Parental communication prevention is mainly attributed to the parent with whom the child resides and “*Greek family courts routinely favor mothers in child custody proceedings even in cases where residence with the father would clearly be in the best interests of the child*” [23].

In most cases (62.0%) there was only a single child in the separated family. Especially in the post-separation period, the relationship between mother and single child has been described as susceptible to develop into a narcissistic and symbiotic link rendering the necessary detachment and achievement of identity of the child more difficult [24,25]. It is also interesting that when there is a single child, the indirect interference more frequently follows, probably because of the aforementioned parent–child relationship’s special features.

Girls and boys were equally involved in cases of parental communication prevention. However, after examining the children’s age, it was observed that the majority of them belonged in the age group of 7–12 years old (40%). School children of this age are more vulnerable to external influence and at the same time they can express their unwillingness to visit one parent or even their hate against him/her. Moreover, they do not have the cognitive development to balance a range of information and exhibit a genuine ambivalence to draw their own conclusions [26,27]. On the other hand, adolescents have achieved a developmental stage and when pressured by loyalty demands from their opposing parents are more able to rebel against parental authority. At this age, they can maintain a consistent stance of anger and they are more likely to make rigid moral judgments of a parent [6].

In our sample, the only parameter that was found to be significantly associated with the judicial decision was the psychological effect on the child. Children/adolescents subjected to parental alienating behaviors have been reported to present both short and long-term psychological consequences, including low self-esteem, anxiety, depression, substance use, suicidal ideation and suicide attempts, educational disabilities, sleep and eating disorders, lack of self-confidence, distress, frustration, and lack of impulse control. Alienated children and adolescents face issues in trusting their own perceptions and feelings, resulting in an uncertain identity, deep insecurity, and inadequate development of independence and individuality [28,29,30,31].

The rest of the parameters (age and gender of the child, gender of the parent preventing the communication, number of children, gender of the child and whether it is the same with the parent preventing the communication, the manner of prevention, the mention of parental alienation) were not associated with the judicial decision. It has been supported that in practice, parental alienation behaviors are connected with female gender before family courts internationally, based on misogynist assessments and assumptions [32]. In their study of family law judgements in the USA, Meier and Dickson found that a father merely alleging parental alienation was 2.3 times as likely as an alleging mother to receive a favorable decision [22]. This represents a statistically significant bias in favor of fathers, which interestingly and fortunately was not found in our study concerning Greek courts.

Switching custody because of alleged alienation has been described in the international literature. Meier and Dickson found that an allegation of alienation was likely to result in switching custody from mothers to fathers in 50% of cases even if the alienation was not upheld [22]. Regarding our cases, a judicial decision that accepts the prevention of parent–child communication may be used as a serious argument from the side of the alienated parent to claim child custody. In Greece, however, the judicial decision about child custody is taken during another trial. That means, that unfortunately, we could not study the consequences of the decision concerning communication prevention on the decision concerning child custody.

Future research should expand on this interesting topic and investigate the relation between communication prevention decision and switching custody in Greece. Moreover, the cases could be directly recorded by the court’s archives to reach a larger and consequently more representative sample. Furthermore, a comparative analysis of the situation before and after the application of the law about shared custody could also be of special interest. Studying the factors that may affect the judicial decisions is of paramount importance as it can contribute to eliminating prejudice and to establishing a more objective and child-centered way of dealing with the matter of custody assignment.

## 5. Conclusions

Parent–child communication prevention is considered a serious and probably widespread phenomenon in Greece given the substantial increase in divorces during the last decade. Direct and indirect methods of interfering with the abovementioned communication are followed by parents. Mothers are more frequently alleged to prevent communication. Judicial decisions in Greece concerning parent–child communication prevention seem to be unaffected by several factors, such as the age and gender of the child, the gender of the parent preventing the communication, the number of children, the child being the same gender as the parent preventing or the parent being prevented from the communication, the way of prevention, and the reference to parental alienation. The present study is only indicative because of its aforementioned limitations and further studies are needed to fully investigate the situation in Greece and help in preventing alienating behaviors between parents, which may have serious consequences on the mental health of children.

## Figures and Tables

**Table 1 healthcare-10-02522-t001:** Test of independence between ways of preventing communication and several demographic and other characteristics.

	Ways of Preventing Communication			
	Direct Interference	Indirect Interference	Both Ways	*p*
Number of children				
1	18 (81.8%)	7 (46.7%)	4 (44.4%)	**0.041 ^‡^**
>1	4 (18.2%)	8 (53.3%)	5 (55.6%)	
Age (years)				
0–6	8 (47.1%)	0 (0.0%)	1 (12.5%)	0.093 ^§^
6.5–12	7 (41.2%)	7 (77.8%)	5 (62.5%)	
12.5+	2 (11.8%)	2 (22.2%)	2 (25.0%)	
Gender				
Male	8 (44.4%)	3 (30.0%)	2 (40.0%)	0.881 ^§^
Female	10 (55.6%)	7 (70.0%)	3 (60.0%)	
Parent preventing the communication				
Father	3 (16.7%)	7 (53.8%)	0 (0.0%)	**0.009 ^§^**
Mother	15 (83.3%)	6 (46.2%)	8 (100.0%)	
Parent/grandparent prevented from communication				
Father	12 (66.7%)	6 (40.0%)	8 (88.9%)	0.083 ^§^
Mother	3 (16.7%)	7 (46.7%)	0 (0.0%)	
Grandparents	3 (16.7%)	2 (13.3%)	1 (11.1%)	
Same gender as the parent preventing the communication				
No	8 (47.1%)	3 (42.9%)	6 (75.0%)	0.418 ^§^
Yes	9 (52.9%)	4 (57.1%)	2 (25.0%)	
Mention of parental alienation				
No	18 (50.0%)	14 (38.9%)	4 (11.1%)	**0.040 ^§^**
Yes	4 (36.4%)	2 (18.2%)	5 (45.5%)	
Psycological effects on children				
Internalization	0 (0.0%)	3 (60.0%)	2 (40.0%)	0.615 ^§^
Feelings concerning relationship with parents	2 (22.2%)	4 (44.4%)	3 (33.3%)	
Decision				
Prevention accepted	15 (60.0%)	4 (16.0%)	6 (24.0%)	**0.028 ^§^**
Prevention not accepted	7 (31.8%)	12 (54.5%)	3 (13.6%)	

Values refer to absolute and relative frequencies and *p*-value of ^‡^ chi-square test (χ^2^) or ^§^ Monte Carlo simulation. Statistically significant values are marked in bold.

**Table 2 healthcare-10-02522-t002:** Test of independence between jury’s decision and several demographic and other characteristics.

	Decision		
	Prevention Accepted	Prevention Not Accepted	*p*
Number of children			
1	18 (58.1%)	13 (41.9%)	0.390 ^†^
>1	8 (44.4%)	10 (55.6%)	
Age (years)			
0–6	7 (70.0%)	3 (30.0%)	0.561 ^§^
6.5–12	10 (50.0%)	10 (50.0%)	
12.5+	3 (42.9%)	4 (57.1%)	
Gender			
Male	10 (62.5%)	6 (37.5%)	0.335 ^†^
Female	9 (45.0%)	11 (55.0%)	
Parent preventing the communication			
Father	6 (50.0%)	6 (50.0%)	0.742 ^†^
Mother	17 (56.7%)	13 (43.3%)	
Parent prevented from communication			
Father	16 (59.3%)	11 (40.7%)	>0.999 ^§^
Mother	6 (54.5%)	5 (45.5%)	
Grandparents	3 (50.0%)	3 (50.0%)	
Same gender as the parent preventing the communication			
No	12 (66.7%)	6 (33.3%)	0.500 ^†^
Yes	9 (52.9%)	8 (47.1%)	
Mention of parental alienation			
No	19 (48.7%)	20 (51.3%)	0.501 ^†^
Yes	7 (63.6%)	4 (36.4%)	
Psychological effects on child			
Internalization	5 (83.3%)	1 (16.7%)	**0.035 ^†^**
Feelings concerning relationship with parents	2 (20.0%)	8 (80.0%)	

Values refer to absolute and relative frequencies and *p*-value of ^†^ Fisher or ^§^ Monte Carlo simulation. Statistically significant values are marked in bold.

## Data Availability

All data generated or analyzed during this study are included in this published article or are available from the corresponding author on reasonable request.

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
