# Peer review of "Decisions of Greek Courts Securing the Right of Parent–Child Communication and Their Determinants"

_healthcare, 2022, doi:10.3390/healthcare10122522_

Round 1

Reviewer 1 Report (New Reviewer)

This article presents data showing that within the sample, mothers are more likely to be accused of interfering with father-child communication than fathers. 

I felt that this article was far too accepting of the idea of parental alienation. There is very little empirical evidence of this syndrome and widespread concerns that it can be used by male abusers as part of coercive control. Please read some of the ongoing work on this issue so that the concept is properly outlined within the article - e.g. Birchall, J., & Choudhry, S. (2022). ‘I was punished for telling the truth’: how allegations of parental alienation are used to silence, sideline and disempower survivors of domestic abuse in family law proceedings, Journal of Gender-Based Violence6(1), 115-131.  https://bristoluniversitypressdigital.com/view/journals/jgbv/6/1/article-p115.xml

and the ongoing work by CEDAW.

The article needs to be clearer about what sort of data is stored on NOMOS and Isokratis. What does it include? Court orders, judgments, submissions, expert evidence? More explanation is needed to understand what evidence the researchers are using. Is this evidence coming from statements made by fathers? Or is it accepted evidence within judgments.  There is some discussion of this point on p8 but it needs to come first to put your findings in context. 

You need to explain to the reader how the Greek courts deal with custody dispute and where these allegations come in. It is very unclear from your description what you are trying to measure. 

I don't understand what you are counting as a decision by the court. This needs to be explained to understand your statistics. On p 4-5 you have a table which records the factors that affect a jury decision. Does this refer to a fact finding on whether the allegation of communication obstruction is true or is the decision about what happens in the custody decision? What is being heard by a  full jury? If it is just a finding of fact, the factors seem nonsensical. Why should the age of a child make it more likely for the court to find that obstruction is happening? 

There is some explanation of your sample on p 7 but really this needs to come earlier in the article so that your findings can be properly understood. 

Where are the children living at the the time of the dispute? The data would obviously be more likely to reflect that mothers are disrupting contact if the children are more likely to live with their mothers post relationship breakdown. 

I don't really understand the value of this study. The data seems limited to suggesting that allegations of obstructing contact are more commonly made by fathers against mothers rather than the other way around. However, children generally live with their mothers post separation in Greece. 

Their might be value in outlining how often parental alienation is coming up and whether this has changed over time. 

I did not understand the point about how the psychological effects on the child of communication obstruction were the most statistically significant for judicial decision making. What is the evidence base for this? Are you arguing that is was the most likely factor in the jury deciding that communication obstruction was actually happening or taking some action on the custody situation like swapping custody or putting in safeguards or parenting supports. 

Author Response

We would like to thank the Editor and the Reviewers for their valuable suggestions and for giving us the opportunity to submit a revised version of our manuscript. We appreciate the Reviewers’ insight, and we hope that we accommodated most of their suggestions in the revised manuscript. All changes regarding English editing have been tracked in the revised manuscript. Changes regarding the content in accordance with reviewers’ comments have been highlighted in yellow.

Reviewer 1

This article presents data showing that within the sample, mothers are more likely to be accused of interfering with father-child communication than fathers. 

I felt that this article was far too accepting of the idea of parental alienation. There is very little empirical evidence of this syndrome and widespread concerns that it can be used by male abusers as part of coercive control. Please read some of the ongoing work on this issue so that the concept is properly outlined within the article - e.g. Birchall, J., & Choudhry, S. (2022). ‘I was punished for telling the truth’: how allegations of parental alienation are used to silence, sideline and disempower survivors of domestic abuse in family law proceedings, Journal of Gender-Based Violence6(1), 115-131.  https://bristoluniversitypressdigital.com/view/journals/jgbv/6/1/article-p115.xml

and the ongoing work by CEDAW.

The following paragraph was added in the Discussion: On the other side, it must be recognized that allegations of parental alienation may frequently be used by abusive fathers to cover and undermine intrafamilial violence and abuse in Courts during child custody arrangements. The concept of parental alienation has been questioned because of the harmful impact it may have on the victims of domestic abuse, mostly mothers and their children [15]. According to the statement of the Experts on Discrimination and Violence against Women (EDVAW) the abuse of the term “Parental Alienation” and of other similar concepts and terms is discouraged, given that this tactic could invoke the denial of child custody to the mother and grant it to a father accused of intimate partner violence in a manner that ignores the possible risks for the child [16].

The article needs to be clearer about what sort of data is stored on NOMOS and Isokratis. What does it include? Court orders, judgments, submissions, expert evidence? More explanation is needed to understand what evidence the researchers are using. Is this evidence coming from statements made by fathers? Or is it accepted evidence within judgments.  There is some discussion of this point on p8 but it needs to come first to put your findings in context. 

In Greece there is no Common Law and the terms “judgement” and “judicial decision” are used as synonyms.

The phrase was modified as follows: “NOMOS” AND “Isokratis” are Greek legal databases that include anonymized cases and important law texts from Greek courts (judicial decisions or otherwise judgements) usually deposited by lawyers who have dealt with them.

You need to explain to the reader how the Greek courts deal with custody dispute and where these allegations come in. It is very unclear from your description what you are trying to measure. 

The following paragraphs were added in the Discussion:

Under Greek law, responsibility of the minor, namely a child under eighteen (18) years of age, constitutes both a right and a responsibility of the parents, who, in principle, must exert it in common. Parental responsibility includes the upbringing / raising, supervision, education, and training of the child, as well as the determination of the child’s residence (Article 1510(1)(2), Greek Civil Code). Parental responsibility is awarded, therefore, to both parents at a child’s birth and will not be withdrawn from one or both, unless under exceptional circumstances. In case of a divorce, if parents do not agree upon the way of exercising responsibility and allocation of specific rights and obligations, the Court shall regulate the above matters. According to Article 1520 of the Greek Civil Code the parent with whom the child does not reside retains the right of personal communication with him/her. Pursuant to the above article, parents do not have the right to prevent the child from communicating with his/her distant in the ascending line relatives, unless there is a serious reason. Grandparents have the same independent right for personal contact with the child (grandchild), even in the event of death of the parent who was their child. Violation of the above right incures penalties, by means of personal detention and/or fines (articles 950 and 947 of the Greek Code of Civil Procedure). In Greece, cases regarding allegations of prevention of the parent-child communication reach the Courts usually after action by one parent.

Under the Greek Law, in case a minor falls victim of domestic violence, if there is a demand by the other parent, a close relative of the child, the Public Prosecutor or even ex officio, the Court can order any proper means to avert the exercise of any form of violence on the child (Greek Law 3500/2006, article 12, 18). These means include the removal of parental responsibility from one parent in total or partly or the offender’s removal from the family residence. The Court can also prohibit him/her to approach the children’s residences or even schools. The same of course accounts for any other kind of abuse, sexual or psychological or neglect [20]. The judicial decisions must be conformed with the standard of the best interest of the child, as this derives from Article 3 of the United Nations Convention on the Rights of the Child [21]. The decision regarding parental responsibility is taken during another trial, simultaneously or not with the trial for the prevention of the parent-child communication.

I don't understand what you are counting as a decision by the court. This needs to be explained to understand your statistics.

In Greece there is no Common Law and the terms “judgement” and “judicial decision” are used as synonyms.

On p 4-5 you have a table which records the factors that affect a jury decision. Does this refer to a fact finding on whether the allegation of communication obstruction is true or is the decision about what happens in the custody decision? What is being heard by a  full jury? If it is just a finding of fact, the factors seem nonsensical. Why should the age of a child make it more likely for the court to find that obstruction is happening? There is some explanation of your sample on p 7 but really this needs to come earlier in the article so that your findings can be properly understood. 

The following phrase was added in the Methods: The judicial decision refers only to the existence or not of the prevention of communication, as the child custody is decided during another trial.

Where are the children living at the time of the dispute? The data would obviously be more likely to reflect that mothers are disrupting contact if the children are more likely to live with their mothers’ post relationship breakdown. 

The following phrase was added in the results: All the allegations were made in the context of separated families and the children were residing mostly with the one parent.

I don't really understand the value of this study. The data seems limited to suggesting that allegations of obstructing contact are more commonly made by fathers against mothers rather than the other way around. However, children generally live with their mothers post separation in Greece. 

There might be value in outlining how often parental alienation is coming up and whether this has changed over time. 

The aim of the present study was to a) discuss the parent-child communication problems, with emphasis in the prevention (interference) of the parent-child communication and parental alienation, b) to refer to the concerns arising in case of child abuse allegation during the trial, c) record cases of parental communication prevention reaching the Greek Courts and determine their characteristics, as well as the factors that may affect the judicial decision.

I did not understand the point about how the psychological effects on the child of communication obstruction were the most statistically significant for judicial decision making. What is the evidence base for this? Are you arguing that is was the most likely factor in the jury deciding that communication obstruction was actually happening or taking some action on the custody situation like swapping custody or putting in safeguards or parenting supports. 

The following phrase was added in the discussion: Regarding our cases, a judicial decision that accepts the prevention of parent-child communication may be used as a serious argument from the side of the alienated parent to claim child custody. In Greece, however, the judicial decision about child custody is taken during another trial. That means, that unfortunately, we did not have the possibility to study the consequences of the decision concerning communication prevention on the decision concerning child custody.

Reviewer 2 Report (New Reviewer)

Dear Authors,

Great paper! Great topic! Worth reading!

One important suggestion, please expand the conclusions regarding the issue of psychological effects on child as the only parametr that is associated with judicial decisions (299–300):

– formal question: indicate which legal principle of Greek law allows the courts to take into account the psychological effects – e.g. protecting the good of children, protecting the parent, protecting other values? etc.

– there were cases of grandparents in your set: were they protected from communication-prevention as being in the place of parents, or as grandparents

– were references to psychological effects considered by one of the parties arbitrary, and contrary to parental rights toward the children?

Please revise English: why do you write judicial with a capital letter? 

Author Response

We would like to thank the Editor and the Reviewers for their valuable suggestions and for giving us the opportunity to submit a revised version of our manuscript. We appreciate the Reviewers’ insight, and we hope that we accommodated most of their suggestions in the revised manuscript. All changes regarding English editing have been tracked in the revised manuscript. Changes regarding the content in accordance with reviewers’ comments have been highlighted in yellow.

Reviewer 2

Great paper! Great topic! Worth reading!

One important suggestion, please expand the conclusions regarding the issue of psychological effects on child as the only parameter that is associated with judicial decisions (299–300):

The following paragraph was added in the discussion: Children/adolescents subjected to parental alienating behaviors have been reported to present both short and long-term psychological consequences, including low self-esteem, anxiety, depression, substance use, suicidal ideation and suicide attempts, educational disabilities, sleep and eating disorders, lack of self-confidence, distress, frustration and lack of impulse control. Alienated children and adolescents face issues in trusting their own perceptions and feelings, resulting in an uncertain identity, deep insecurity and inadequate development of independence and individuality [28-31].

formal question: indicate which legal principle of Greek law allows the courts to take into account the psychological effects – e.g. protecting the good of children, protecting the parent, protecting other values? etc.

The following phrase was added in the discussion: The judicial decisions must be conformed with the standard of the best interest of the child, as this derives from Article 3 of the United Nations Convention on the Rights of the Child.

there were cases of grandparents in your set: were they protected from communication-prevention as being in the place of parents, or as grandparents

In the table 1 “parents/grandparents prevented from communication” was used instead of “parents prevented from communication”

– were references to psychological effects considered by one of the parties arbitrary, and contrary to parental rights toward the children?

The following phrase was added in the Methods: The description of the psychological effects in the judicial decision was based on the evidence assessed by the testimonies and the report of the child psychiatrist or psychologist (if performed).

Please revise English: why do you write judicial with a capital letter? 

English language editing has been performed. «Judicial» has also replaced by «judicial».

Round 2

Reviewer 1 Report (New Reviewer)

Hi, 

I appreciate that Greece is not a common law jurisdiction and decisions and judgments are used interchangably but for and international audience you need to be clearer on what your data is based on.  

Can you just explain in one sentence what you based your finding on? Are they written rulings from the court on particular? You say that data is uploaded onto to the databases by pracitioners. Does this mean that the sample is a bit sporadic?  

Author Response

We would like to thank the Editor and the Reviewer for their valuable suggestions and for giving us the opportunity to submit a revised version of our manuscript. We appreciate the Reviewers’ insight, and we hope that we accommodated most of their suggestions in the revised manuscript. All changes regarding English editing have been tracked in the revised manuscript.

I appreciate that Greece is not a common law jurisdiction and decisions and judgments are used interchangeably but for and international audience you need to be clearer on what your data is based on.  Can you just explain in one sentence what you based your finding on? Are they written rulings from the court on particular?

We modified the paragraph in the Methods section regarding the law texts in the Greek legal databases as follows:

“The Greek legal databases “NOMOS” and “Isokratis” include anonymized cases and important law texts from Greek courts, usually deposited by lawyers who have dealt with them. Cases included in the above databases were chosen in an almost random way and thus are rather indicative of the situation in Greece. The law texts include final judicial decisions, in which the reasons why the court reached its conclusion are also included. It is worth mentioning that Greece is a civil law country, which means that, unlike Anglo-American common law, the judicial decisions are based only on enacted laws (in the form of codes or other statutes) (Greek Civil Code, Art. 1) and international law.”

You say that data is uploaded onto to the databases by practitioners. Does this mean that the sample is a bit sporadic?  

Regarding the question (if the sample is a bit sporadic) we comment in the Methods and in the Discussion that: “Cases included in the above databases were chosen in an almost random way and thus are rather indicative of the situation in Greece.” (Methods) and “Obviously, not all the cases are included in the above databases, and even in the existing ones many details are lacking due to Personal Data Protection Act. These are two methodological limitations we should consider prior to interpreting the findings and drawing conclusions” (Discussion). Finally, we highlight the need for further research: “Moreover, the cases could be directly recorded by the Court’s archives to reach a larger and consequently more representative sample” (Discussion).

This manuscript is a resubmission of an earlier submission. The following is a list of the peer review reports and author responses from that submission.

Round 1

Reviewer 1 Report

Some suggestions for changes to the article:

1/ The title:  Dealing With Cases Of Parent-Child Communication Prevention In Greek Courts: Parental Alienation Or Child Abuse?

The title of the article is long and raises doubts - it is worth considering the possibilities of shortening or modifying it.

for example:

Greek court rulings ensuring the communication / contact rights of (both) divorced parents and the child and their conditions.

Decisions of Greek courts regarding violation of the right of communication / contact of (both) divorced parents with the child and factors influencing these decisions / court decisions.

Decisions of courts in Greece securing the right of communication / contact of (both) divorced parents with the child and their determinants.

2/ It would be justified to rethink the structure or content of some tasks in the text, because the reader may have difficulty understanding them.

For example:

Analysis of the sentence

“Aim of this study is to present the current legal framework in Greece concerning parental alienation and parental communication prevention, as well as to determine how Greek Courts deal with the above cases, which factors may affect the Judicial decisions and if there are specific arrangements when child abuse accusations arise during 50 the trial.”

Excerpt: "Aim of this study is to present the current legal framework in Greece concerning parental alienation and parental communication prevention, ...." - raises the question: is it about (1) preventing communication between parents (parent-parent), or rather about

  (2) preventing the child from restricting communication with one parent - by the other parent? My guess is that the authors mean the second option (2), but I'm not sure if my guess is correct.

Another sentence that is not clear cut:

“The prevalence of conflicting divorces and the consequent hostile behavior between ex-spouses in most of the times lead to disorders of the mental health of children, who accept the consequences of their parents' vengeful attitude.”

  • Is it that - the child accepts the consequences of the hostility between his parents - or
  • is it that - the child bears the consequences of this hostile relationship between his parents? or
  • is it that - the child accepts the hostile attitude of the parent under whose direct care he or she remains - towards the other parent and the child also has a hostile attitude towards this parent (this would mean "narrowing down" the problem to the PAS syndrome) (?)

Next question: How to understand: “the occurrence of conflicting divorces” (?) - is the beginning of this sentence necessary?

Next:

“Except from factors related to parents, specific traits (parenting practices, personality, mental health e.t.c.) and their relationship before and after separation, child factors (age, cognitive capacity, temperament, vulnerability, special needs, and resilience), and the  course of the adversarial process/litigation have also been recognized as dynamic contributing and facilitating factors (3).”

I suggest moderate change: I propose a little change: removing the words “specific traits”.

It would be beneficial for the text if the Authors carefully analyze all their considerations and the thoughts / ideas they intend to express: section by section, sentence by sentence.

I assumed that some "ambiguities" of the text result not from the incompetence of the Authors or the superficial approach to the problem, but from imprecise translation.

In my review, I focus on the positive "points" of the text, namely the importance of the problem and an attempt to combine the psychological and legal perspectives.

I assume that this translation results in many inaccuracies and ambiguities in the authors' deliberations - however, if the text were to be published, it is necessary that the authors take into account the Greek content, and then - the quality of the translation.

….

One of the positives is the synthetic description of the PAS syndrome and the indication of difficulties in its diagnosis.

On the basis of section 1.1. - one would expect that the procedure for diagnosing PAS at the request of a Greek court would be briefly outlined later in the article. Providing such knowledge  - in next part of this article (in the part where the authors present the results of the analyzes) - would be very useful for psychologists (from different countries) who, at the request of the court, undertake the diagnosis of a child's family situation and psychosocial (and somatic health) functioning. Hence the request to the Authors to enter such information in the text.

Unfortunately, I do not know if the authors collected such data - do they have access to it (?)

In point 2.

The authors indicated a method of data collection that is not characteristic of psychological research - it would be justified to provide a brief explanation of this methodology. In the earlier point (1.2), the authors presented information on the differentiation of difficulties in child-parent communication - depending on the gender of the parent entrusted with the care of the child by the court. It would be justified to extend this point to the other factors indicated below (in point 2) - based on the data from the literature review (psychological, sociological, family law?)

Factors (listed in section 2) extracted from the database were the following - in addition to the parent's gender - number of children and their gender, communication prevention strategies used, court order (acceptance or lack of communication), reference to the term "parental alienation" -  hence the request for a brief discussion of these factors on the basis of specialist literature (for example in p. 1.2).

A brief explanation of Direct interference and Indirect interference is also justified - how was it classified? Was the classification / categorization made by competent judges? Was the description of these strategies included in the forensic material?

It would be good to indicate the limitations of the research presented and the directions of future research on the problem discussed in this article.

In the last part of the article, the authors wrote:

“The aim of the present study was to a) discuss the parent-child communication prob-386 lems, with emphasis in the prevention (interference) of the communication and parental 387 alienation, b) to refer to the concerns arising in case of child abuse allegation during the  trial, c) record cases of parental communication prevention reaching the Greek Courts and 389 determine their characteristics, as well as the factors that may affect the judicial decision 390 and d) describe the legal framework and the current situation in Greece on the matter.”

However, it is worth adding that the goals were partially achieved - on the basis of a review of specialist literature (the focus is on the PAS syndrome) - and partially in the course of empirical research.

The article - as I have already emphasized - concerns a very important problem from the perspective of individual and social development. In my opinion, the text is noteworthy and can be interesting and useful for psychologists, sociologists, educators and lawyers. It is worth bringing these issues closer to the so-called "society".

I suggest introducing the indicated modifications and carefully analyzing the text and the translation.

This is necessary, although it can be difficult due to the "dual" structure of the text - which includes psychological and legal considerations. However, this combination is justified due to the problem raised in the article.

Reviewer 2 Report

major comments
The article entitled "Addressing Parent-Child Communication Prevention Cases in Greek Courts: Parental Alienation or Child Abuse?" addresses the issue of parental alienation.
Unfortunately, the article appears unclear and inconsistent between introduction, objectives, method, and conclusion.

The title
The title is misleading with respect to the content. It is also unclear whether the researchers want to show that parental alienation does not exist ,but that it is a child abuse problem, or parental alienation hides other forms of abuse.

introduction
- Although the authors at the beginning explain quite well the evolution that the PAS has had, especially the impossibility of framing it as a syndrome, they do not link the psychological research conducted so far with the objectives of their work.
- The legislative framework presented, although useful for understanding the reference culture, appears unbalanced compared to the other parts of the article. I would have expected more discussions on the state of the art of psychological research done to date.

Some methodological problems:
- The authors do not clarify how the data were extracted or which content analysis methodology was used (inclusion / exclusion criteria)
- It is not clear on which scientific premises and / or hypotheses the researchers expected associations between the jury's decision and the different variables (sex, age, etc.)
- Researchers make a distinction between direct and indirect interference as a means of preventing communication, but it is not clear how we distinguished the two categories

Results
the data, although they are a vision of the Greek situation, are not adequately discussed in the light of the reference literature

conclusions
the conclusions are very concise and vague and do not highlight the advancement of knowledge that is given to this topic by their research

minor comments

there are many typos and errors

for example

the article "the" is always accented

line 122 causai 

line 207 legai

line 205 cruciai

Reviewer 3 Report

This manuscript reports on a study conducted of legal cases in Greece that looked at behaviors of parents involved in legal cases where parental alienation had been labeled as an issue. While there are some interesting findings, there are a large number of conceptual and methodological concerns that need to be addressed before I would recommend it being considered again for publication.

  1. I understand and am very empathetic to the challenges of writing a scholarly paper in a language that is not the author’s primary language, and a lot of concerns I have may stem from how particular constructs were being described by the authors. I strongly recommend that the authors seek someone whose primary/first language is English to help address many grammatical issues throughout the paper. There can also be refinement in the structure of the paper—for example, there were some paragraphs that only contained one sentence (e.g., see line 342). In this area of research, precise terminology is important. Terminology concerns are addressed specifically below.
  2. The use of the word “followed” (e.g., see the abstract) is not clear. Do the authors mean perpetrated?
  3. On lines 35-36, it is important to include the more commonly accepted terminology of PA today…much of the paper seems focused on early Gardner definitions, and there have been considerable advances since then. The authors are encouraged to find reviews of the recent scientific literature in books such as Lorandos & Bernet (2020), Parental Alienation-Science and Law, or even the Harman, Kruk & Hines (2018) paper that is referenced. For example, the authors did not mention that an important part of the definition of PA is a child’s alignment with a preferred parent, alongside the rejection of their other parent for reasons that are not legitimate.
  4. There appears to be a heavy reliance in the paper on “critics” of PA theory who have not published or referenced much scientific or empirical research support their opinions. I am not sure whether the authors have read the critiques of the work of the people that were cited, but they are clearly omitted from this paper, which then presents a biased and one-sided perspective on the problem. Examples of this are below, as well as where the authors are encouraged to look to see the problems with the citations that were used to support their arguments.
    1. On line 44, the authors cite Milchman (2019) and state that parental alienation seems to be frequently used in cases of alleged child abuse. Yet, there are many other scholars who have noted that this is not the case. For example, in cases where parental alienation was alleged or found to have occurred, less than half cases had any allegation of abuse at all (including IPV allegations; see Harman & Lorandos, 2020). Indeed, even Dr. Gardner wrote about this in his original writings: that allegations of child abuse occur in only a small portion of PA cases.
    2. In the same sentence, the authors cite Milchman to claim that scientific tools have been inadequate to discriminate parental alienation. I am not sure if the authors are unaware of the greater scientific literature on the topic of PA, or advances in differentiating cases of PA, but there are now several papers documenting how PA is reliably differentiated from other forms of family conflict using the five-factor model (see Baker’s original four-factor model, 2020; Bernet, 2020; and Lorandos & Bernet, 2020). Other opinions should be referenced and discussed.
    3. The authors cite Meier & Dickson in line 402 that 82% of claims of PA are made by fathers. Yet, the sampling issues of the Meier data make this unreliable (see Harman & Lorandos, 2020 critique of the study on which the claim is based). Indeed, in the Meier et al., 2019 paper, the authors explicitly stated on page 8, “The PI and consultant Dickson developed analyses for the statistical consultant to complete, reviewed the output, and, through numerous iterations, refined, corrected, and amplified on the particular analyses.” This statement indicates that the authors manipulated their data (otherwise known as p-hacking) to get the results they wanted…a highly unethical practice and makes any conclusions from the study unreliable. That study was the basis of the Meier & Dickson paper. I would carefully review the references used throughout the paper that are contradicted by data reported by other authors and do a critical analysis of the methods used by the authors, before citing them in support of the arguments being made in the paper. I encourage you to look to Lorandos, 2020 for a paper discussing gender differences in appellate cases, and to the more recent prevalence studies on PA that have been published over the last few years.
    4. On line 432, the authors cite Milchman and “misogynist” decisions, as well as Meier & Dickson again. See 4c above. Harman and Lorandos (2020) did not find gender differences in how courts made decisions about PA.
  5. The authors use the term “parental communication interference” and do not reference the larger literature or research on parental alienating behaviors (e.g., see Baker & Darnall, 2006; Harman, Kruk & Hines, 2018; Harman & Matthewson, 2020). It seems the authors are talking about gatekeeping behaviors that have their own line of research and do not always even reference parental alienation (e.g., see work by William Austin). Communication interference implies that the behaviors are related to blocking phone calls, texts, or visits. Yet in the results section, it seems that the authors included other parental alienating behaviors (e.g., tension during contacts) under this category without any explanation why.
  6. It is unclear what the authors mean by “direct” and “indirect” communication interference, and why they examined these separately. No theoretical justification was provided. The authors are referred to Harman, Lorandos, Grubb & Biringen (2020) for examples of direct/indirect aggression and why this may matter theoretically in this context.
  7. On line 53, the authors state that in “most cases,” hostile behavior between ex-spouses leads to “disorders of mental health” of children. It is unclear what the basis of this statement is. Not all children exposed to parental alienating behaviors become alienated (in fact, a small % of them actually do; see Harman, Leder-Elder, & Biringen, 2019). Likewise, some children are very resilient and able to adapt, even in other very difficult environments. A blanket statement that “most” children develop disorders needs to be supported with reliably acquired statistics.
  8. On line 69, the authors cite a case where there was “realistic alienation” due a father’s behaviors. Terminology is important here, as the situation described is not parental alienation, but parental estrangement.
  9. On line 122, the authors cite a 2003 paper by Johnston (who is female, not male) who has claimed that PA is multifactorial, and not just caused by an alienating parent. This old paper has been critiqued heavily by scholars, including Gardner himself, and there is no indication the authors considered the criticisms of her approach. I recommend the authors refer to newer papers such as Bernet, Wamboldt, & Narrows, 2016; Bernet, 2020; and the Lorandos & Bernet 2020 book for more updated discussions about the factors that cause different forms of family conflicts. For example, loyalty conflicts describe situations where both parents contribute to the problem and the child is triangulated. This is a very different family dynamic than parental alienation where the power dynamics are comparatively more asymmetrical and the alienating parent engages in coercively controlling behaviors. Blaming both parents for PA neglects how different forms of IPV operate (e.g., situational couple violence versus intimate terrorism/coercive control)---we would not blame victims of battery for their abuse. There has been substantial evidence accumulating the last few years to indicate that PA not only resembles coercive control, but they are two sides of the same coin (e.g., see Joshi, 2020).
  10. The authors state on line 144 that mothers are more effective in alienating children than fathers based on one study (the 2003 Johnston study). Population based studies indicate that there are NOT gender differences in who is more likely to be an alienated parent (e.g., see Harman et al., 2019), and other studies in Spain and Australia note the same (e.g., see Kaspiew, 2007; Lavadera, 2012 and 2017). Gender differences are found when cases go to court (see Lorandos, 2020), yet so far we only have empirical evidence of this at the appellate level in the U.S. (e.g., see Harman & Lorandos, 2020)
  11. In the paragraph starting on line 180, the authors state that PAS has been used in cases where there is alleged sexual and child abuse. I refer the authors to 4a above. There have been noted sampling issues with scholars who have claimed that all claims of PA are used to get out of child/sex abuse claims (e.g., see the Harman & Lorandos, 2020 critique of Meier et al., 2019). Indeed, of 967 appellate cases, only 77 had a finding of abuse regarding the parent alleged or found to have been alienated from their child. And according to Harman & Lorandos, 2020, the “finding” was liberally applied, as in many cases the findings were later dismissed or found to be false/unsubstantiated. So, it is VERY misleading to state that PAS claims are involved in cases where child abuse and sexual abuse are alleged, as there is no evidence about how often that really happens in well-conducted studies using unbiased sampling strategies.
  12. In the methods section, the authors should more thoroughly describe the search process. What were the Greek words (and their English translations) used? This is important from a replicability perspective. How many cases were derived from the initial search terms? How were cases included/excluded in the database? In the discussion, the authors mention that there were data privacy issues, etc., so it is very unclear what the sample is. Can the data be generalized, and to who?
  13. I struggled a lot with understanding the results section, and it may be translation issues combined with lack of clarity in terminology. For example, line 358 mentions “children’s number,” but I do not know what that means. Does that mean the number of children in the family? Also, on line 366, the authors state that both forms of direct and indirect behaviors were used “exclusively” by mothers, but the results indicated that fathers did the behaviors too.
  14. There has been a recent change in the law about parenting time in Greece, and it would be useful to discuss what the impact may be on families.
  15. I do not know what the authors mean by prevention being “accepted” by the court. I think more details about how the variables were defined, and why they were operationalized the way that they were would be useful.
  16. Given there were not any specific hypotheses or research questions detailed in the study, I found the results and discussion hard to follow. While descriptive studies are useful to obtain basic information about a phenomenon, it would have been more useful to have a set of research questions specified at the start, and then organize the results and discussion around them.

I hope that this feedback useful. I encourage the authors to pursue this work, as it is important to document what is occurring in courts around the world. That said, the research literature cited is not really updated or thorough and presents a gender biased view on the topic that is not supported by other important empirical work.